# Entropy-Based Estimation of Event-Related De/Synchronization in Motor Imagery Using Vector-Quantized Patterns

**DOI:** 10.3390/e22060703

**Published:** 2020-06-24

**Authors:** Luisa Velasquez-Martinez, Julián Caicedo-Acosta, Germán Castellanos-Dominguez

**Affiliations:** Signal Processing and Recognition Group, Universidad Nacional de Colombia, Manizales 170004, Colombia; juccaicedoac@unal.edu.co (J.C.-A.); cgcastellanosd@unal.edu.co (G.C.-D.)

**Keywords:** event-related de/synchronization, entropy, motor imagery, vector quantization

## Abstract

Assessment of brain dynamics elicited by motor imagery (MI) tasks contributes to clinical and learning applications. In this regard, Event-Related Desynchronization/Synchronization (ERD/S) is computed from Electroencephalographic signals, which show considerable variations in complexity. We present an Entropy-based method, termed *VQEnt*, for estimation of ERD/S using quantized stochastic patterns as a symbolic space, aiming to improve their discriminability and physiological interpretability. The proposed method builds the probabilistic priors by assessing the Gaussian similarity between the input measured data and their reduced vector-quantized representation. The validating results of a bi-class imagine task database (left and right hand) prove that *VQEnt* holds symbols that encode several neighboring samples, providing similar or even better accuracy than the other baseline sample-based algorithms of Entropy estimation. Besides, the performed ERD/S time-series are close enough to the trajectories extracted by the variational percentage of EEG signal power and fulfill the physiological MI paradigm. In BCI literate individuals, the *VQEnt* estimator presents the most accurate outcomes at a lower amount of electrodes placed in the sensorimotor cortex so that reduced channel set directly involved with the MI paradigm is enough to discriminate between tasks, providing an accuracy similar to the performed by the whole electrode set.

## 1. Introduction

The Motor Imagery (MI) paradigm is a class of Brain-Computer Interfaces (BCI) that performs the imagination of a motor action without any real execution, relying on the similarities between imagined and executed actions at the neural level. Understanding of MI fundamentals gives insights into the underpinning brain dynamic organization since a mental representation of specific movements involves cooperating (sub-)cortical networks in the brain. Thus, evaluation and interpretation of brain dynamics in the sensorimotor area may contribute to the assessment of pathological conditions, the rehabilitation of motor functions [1,2], motor learning and performance [3], evaluation of brain activity functioning in children with developmental coordination disorders [4], improving balance and mobility outcomes in older adults [5], and more recently in education scenarios, allows analyzing learner’s mental situation under frameworks as the Media and Information Literacy methodology [6] and John Sweller’s Cognitive LoadTheory [7], among other applications.

Elicited by MI activity, Event-Related Desynchronization/Synchronization (ERD/S) is computed from Electroencephalographic signals (EEG) to capture channel-wise temporal dynamics related to both sensory and cognitive processes. So, ERD/S is a time-locked change of ongoing EEG signals of electrodes placed in the sensorimotor area, showing an intensified cooperation between the decreasing ipsilateral and increased contralateral motor regions for movement representations. Conventionally, ERD/S is estimated by the instantaneous amplitude power that is normalized to a reference-time level and averaged over a representative amount of EEG trails in an attempt to improve the signal-to-noise ratio [8]. For decreasing the inherent inter-subject variability, the scatters of trial power must be accurately reduced, usually by a trial-and-error procedure, hindering the detection and classification of motor-related patterns in single-trial training. Correcting the baseline of each single-trial before averaging spectral estimates is an alternative method [9]. Nonetheless, the ERD/S patterns are characterized by its fairly localized topography and frequency specificity, making this approach include a priori choice of frequency bands. However, the band-passed oscillatory responses tend to depreciate a wide range of nonlinear and non-stationary dynamics, which may be interacting in response to a given stimulus by synchronization of oscillatory activities [10].

As a consequence of the nonstationarity and nonlinearity of acquired EEG data [11], the MI brain activity shows considerable variations in complexity of the physiological system with dynamics affected by motor tasks that can be perceived in the pre-stimulus activity and the elicited responses. Thus, the extracted ERS/D time-courses can be modeled as the output of a nonlinear system. In this regard, various measures are reported to quantify the complex dynamics of elicited brain activity, like Kolmogorov complexity [12], Permutation Entropy, Sample Entropy, and its derived modification termed Fuzzy Entropy [13] that provides a fuzzy boundary for similarity measurements [14], or even the fusion of Entropy estimators to achieve the complementarity among different features, as developed in [15]. However, extraction of ERD/S dynamics using Entropy-based pattern estimation is hampered by several factors like movement artifacts during recording, temporal stability of mirroring activation over several sessions differs notably between MI time intervals [16], low EEG signal-to-noise ratio, poor performance in small-sample setting [17], and inter-subject variability in EEG Dynamics [18]. Hence, the reliability of Entropy-based estimators may be limited by several factors like lacking continuity, robustness to noise, and biasing derived from superimposed trends in signals.

One approach to yield more statistical stability from Sample-based estimators is to transform the time series into a symbolic space, from which the regularity of MI activity is measured like in the case of Permutation Entropy that associates each time series with a probability distribution, whose elements are the frequencies connected with feasible permutation patterns, and being computationally fast [19]. Since the irregularity indicator considers only the order of amplitude values, several variations to the initially developed permutation Entropy are proposed to tackle the problem of information discarding. Thus, Dispersion Entropy appraises the frequency of a symbolic space that is built in mapping each sample through a class pattern set across epochs [20], retaining higher sensitivity to amplitude differences and accepting adjacent instances of the same class [21]. Further improvements can be achieved by introducing information about amplitudes and distances [22,23]. Besides entering more free parameters to tune, the sample-based estimators face additional restrictions in the extraction of ERD/S dynamics like the fact that motor imagery activity reduces the EEG signal complexity [24]. Also, there is a need for a careful choice of the time window that mostly affects the effectiveness of short-time feature extraction procedures; it must have enough length to cover the interval within a neural pattern is activated, while at the same time it should remove the unrelated sampling points [25].

Here, we present an Entropy-based estimation of ERD/S using quantized stochastic patterns as symbolic space, aiming to improve the discriminability and physiological interpretability of motor imagery tasks. The proposed Entropy-based estimator, termed *VQEnt*, is sample-based that builds the probabilistic priors by assessing the Gaussian similarity between the input and its reduced vector-quantized representation to extract more information about amplitudes of time-courses. The validating results, obtained on the widely used database of bi-class imagine tasks, (left and right hand) show that *VQEnt* holds symbols that encode several neighboring samples, providing similar or even better accuracy than the other baseline sample-based algorithms of Entropy estimation. Moreover, the performed ERD/S time-series are close enough to the trajectories extracted by the variational percentage in EEG signal power regarding a reference interval, fulfilling the physiological of the MI paradigm. In the case of individuals with BCI literacy, the *VQEnt* estimator presents the most accurate outcomes at a lower amount of electrodes placed in the sensorimotor cortex so that reduced channel set directly involved with the MI paradigm is enough to discriminate between tasks, providing an accuracy similar to the performed by the whole electrode set. The agenda is as follows: Section 2 describes the collection of MI data used for validation. It also presents the fundamentals of complexity-based estimation of time-evolving ERD/S and describes the used quantized stochastic patterns, defining the required probabilistic priors for similarity-based calculation. Further, Section 3 provides a summary of the results for evaluating the interpretation of ERD/S as well as their contribution to distinguishing between MI tasks. Lastly, Section 4 gives critical insights into their supplied performance, and address some limitations and possibilities of the presented approach.

## 2. Materials and Methods

### 2.1. EEG Recordings and Preprocessing

The proposed entropy-based approach for ERP estimation is evaluated experimentally on a public collection of EEG signals recorded in a 22-electrode montage from nine subjects (BCI competition IV dataset IIa (http://www.bbci.de/competition/iv/)). The dataset was collected in six runs separated by short breaks. Each run contained 48 trials lasting 7 s and distributed as depicted in Figure 1. To perform each MI task (left and right hand with labels noted as $\lambda \in \{l,l^{\prime }\}$, respectively), a short beep indicated the trial beginning followed by a fixation cross that appeared on the black screen within the first 2 s. Next, as the cue, an arrow (pointing to the left, right, up or down) appeared during 1.25 s, indicating the specific MI task to imagine regarding one of four MI tasks, i.e., left hand, right hand, both feet, and tongue, respectively. Then, each subject performed the demanded MI task while the cross re-appeared in the following time interval (MI segment), ranging from 3.25 to 6 s.

The preprocessing EEG stage comprises data filtering, segmentation of MI intervals, and data referencing since we only validate the labeled trials, having removed artifacts provided by the database. Initially, for selecting the discriminant information of MI responses, each raw EEG channel $x^{c}\in R^{T}$ is sampled at 250 Hz (i.e., at sample rate Δt=0.004 s) and passed through a five-order bandpass Butterworth filter within Ω=[4,40] Hz. Afterwards, the MI time window $T_{MI}=2$ s is segmented. Then, we deal with the volume conduction effect that produces a low signal-to-noise ratio of EEG data by applying the Laplacian spatial filter [26]. The preprocessing procedures are implemented using a tailor-made software in Phyton.

### 2.2. Complexity-Based Estimation of Time-Evolving Event-Related De/Synchronization (ERD/S)

This time-locked change of ongoing EEG is a control-mechanism of the somatotopically organized areas of the primary motor cortex, which can be generated intentionally by mental imagery. For each measured EEG recording $x_{n}\in \left[x_{\Delta t,n}\in R\right]$, the estimation of ERD/S is performed, at specific and sample Δt∈T, by squaring of samples and averaging over the EEG trial set to compute the variational percentage (decrease or increase) in EEG signal power regarding a reference interval as follows [27]:
(1a)$$\begin{array}{cc} \zeta_{\Delta t}^{P} & =\left(\xi_{\Delta t}-\bar{\xi }\right)/\bar{\xi } \end{array}$$
(1b)$$\begin{array}{cc} s.t.: & \mathrm{var}\left(\xi_{\Delta t}\right)\gg \mathrm{var}\left(\bar{\xi }\right) \end{array}$$
where $\xi_{\Delta t}=E\{|x_{\Delta t,n}{|}^{2}\in x_{n}:\forall n\}$ is the power scatter averaged across the trial set, n∈N, and the trial power scatter $\bar{\xi }=E\left\{\xi_{\Delta t}:\forall \Delta t\in \Delta T_{0}\right\}$, with $\bar{\xi }\in R$, is computed by averaging over the reference time interval $\Delta T_{0}\subset T$, being $T\in R^{+}$ the whole EEG recording segment. Due to each time-series of ERD/S is computed across the whole trial set, the inherent inter-subject variability implies to fulfill the restriction Equation (1b) by ruling accurately the trial power scatter $\bar{\xi }\left(\cdot \right)$.

Instead of using the power-based estimates in Equation ([Disp-formula FD1a-entropy-22-00703]) that are assessed across the trial set, the ERD/S time series can be computed in a one-trial version, for instance, by measuring the Entropy of time-series changes over time as below:
(2a)$$\begin{array}{cc} \zeta_{\Delta t}^{H} & =E\{H\left\{X_{n}\left(\tau \right)\right\}:\tau \in T\},\;\tau >\Delta t \end{array}$$
(2b)$$\begin{array}{cc} s.t.: & |\partial H\{X_{n}\left(\tau \right)\}/\partial \tau |\mathrm{exists}\mathrm{for}\mathrm{every}\tau \subset T \end{array}$$
where $X_{n}\left(\cdot \right)$ are the state-space partition sets that can be extracted within a time window lasting $\tau =N_{\tau }\Delta t$. In terms of the Entropy metric H{·}, the newly-introduced restriction Equation (2b) relies upon the assumption that several samples might be compared to itself when two consecutive time windows commonly consist of the same samples. So, the discrete-time space-state models can be built in the form of a following embedded representation: (3)$$X_{n}\left(\tau ,M\right)=\left\{{\tilde{x}}_{n}\left(\tau ,M;q\right)=\left[x_{m\Delta t,n}\left(\tau ;q\right):m\in \left[q,q+M-1\right]\right]:q\in Q\right\},\;Q=N_{\tau }-M$$
where *M* is the embedding dimension, *Q* is the size of the state-space or alphabet, and $\left\{{\tilde{x}}_{n}\left(\cdot ,\cdot ;q\right)\in R^{M}\right\}$ is the windowed representation or symbol.

Thus, the Entropy in Equation (2a) can be estimated at a time window τ by a pairwise comparison between a couple of embedded versions $\pi_{n}\left(\cdot ,\rho ;\tau \right)$: (4)$$H\left\{X_{n}\left(\tau ,M\right);\rho \right\}=-\ln\left(\pi_{n}\left(M+1,\rho ;\tau \right)/\pi_{n}\left(M,\rho ;\tau \right)\right)$$

Relying on the fact that π(·,·;·) is the probability that two sequences are similar within *M* points, a direct calculation is through the mean value of pattern count that is evaluated as: (5)$$\pi_{n}\left(M,\rho ;\tau \right)=E\left\{\mathrm{num}\left\{d\left({\tilde{x}}_{n}\left(\tau ,M;q\right),{\tilde{x}}_{n}\left(\tau ,M;q^{\prime }\right)\right)<\rho \right\}:\forall q,q^{\prime }\in Q,q\neq q^{\prime }\right\}$$
where num{d(·,·)}∈N is the count of distance lower than tolerance $\rho \in R^{+}$, $d\left(\cdot ,\cdot \right)\in R^{+}$ is the distance between a couple of embedded partitions. So, two widely-known distances are used [28]:
(6a)$$\begin{array}{cc} \mathrm{SampEnt}:\; & d_{S}\left({\tilde{x}}_{n}\left(\tau ,M;q\right),{\tilde{x}}_{n}\left(\tau ,M;q^{\prime }\right)\right)=\max_{\bar{\forall m\in M}}\left|x_{m\Delta t,n}\left(\tau ;q\right)-x_{m\Delta t,n}\left(\tau ;q^{\prime }\right)\right| \end{array}$$
(6b)$$\begin{array}{cc} \mathrm{FuzzyEnt}:\; & d_{F}\left({\tilde{x}}_{n}\left(\tau ,M;q\right),{\tilde{x}}_{n}\left(\tau ,M;q^{\prime }\right)\right)=\exp\left(d_{S}{\left({\tilde{x}}_{n}\left(\tau ,M;q\right),{\tilde{x}}_{n}\left(\tau ,M;q^{\prime }\right)\right)}^{2}/\rho \right) \end{array}$$
where $\rho ∼0.1\sigma_{\tilde{x}}$ and $\sigma_{\tilde{x}}$ is the standard deviation of the measured EEG data.

### 2.3. Symbolic Spaces Using Quantized Stochastic Patterns

The pattern count in  Equation (Equation 4) can be alternatively assessed through the conditional probability that two stochastic models, extracted from the same embedded representation in  Equation (Equation 3), are similar [29]. In particular, we estimate the conditional probability $p\left({\bar{x}}_{n}\left(\tau ,M;\cdot \right)|X_{n}\left(\tau ,M\right)\right)$ that reflects the closeness between the original expanded state-space partition set, $X_{n}\left(\cdot ,\cdot \right)$ and every element of an equivalent stochastic representation with reduced dimension, ${\bar{x}}_{n}\left(\cdot ,\cdot ;\cdot \right)\in R^{M}$, created from the original set. Thus, we rewrite the Entropy-based estimation, performed within τ, as below:(7)$$\begin{array}{cc} H\{X_{n}\left(\tau ,M\right);\rho \} & =H\{\left({\bar{x}}_{n}\left(\tau ,M;q^{\prime }\right)|X_{n}\left(\tau ,M\right)\right)\}\\ \; & =-\sum_{q^{\prime }\in Q^{\prime }}p\left({\bar{x}}_{n}\left(\tau ,M;q^{\prime }\right)|X_{n}\left(\tau ,M\right)\right)\log p\left({\bar{x}}_{n}\left(\tau ,M;q^{\prime }\right)|X_{n}\left(\tau ,M\right)\right), \end{array}$$
where the reduced set holds $Q^{\prime }\leq Q$ symbols ${\bar{x}}_{n}\in {\bar{X}}_{n}\left(\cdot ,\cdot \right)$, which are assumed to be more distinct across the whole embedded representation.

We model the alternative embedded set, noted as ${\bar{X}}_{n}\left(\cdot ,\cdot \right)\in R^{Q^{\prime }\times M}$, using quantization techniques, which compress a larger dataset in  Equation (Equation 3) into one smaller equivalent set of code vectors. In particular, we employ the approach described in [30] that finds the closest code-vector representation.

Nevertheless, the similarity pattern count calculation in  Equation (Equation 5) will necessitate more statistics due to the reduced size of the newly introduced embedding stochastic set. Instead, we propose to build the probabilistic priors in  Equation (Equation 4) between both representations (original and VQ-reduced) by calculating the conditional probability that a sample of the unfolded EEG signal belongs to every formed VQ symbol. So, according to Bayes theorem, we have: $$p\left({\bar{x}}_{n}\left(\tau ,M;q^{\prime }\right)|X_{n}\left(\tau ,M\right)\right)=p\left(X_{n}\left(\tau ,M\right)|{\bar{x}}_{n}\left(\tau ,M;q^{\prime }\right)\right)p\left({\bar{x}}_{n}\left(\tau ,M;q^{\prime }\right)\right)$$

Assuming that the input samples follow a Gaussian distribution, we employ the similarity-based approach between sets for estimation of both probabilistic terms, as proposed in [31]:
(8a)$$\begin{array}{cc} p\left(X_{n}\left(\tau ,M\right)|{\bar{x}}_{n}\left(\tau ,M;q^{\prime }\right)\right)∼N\left(X_{n}\left(\tau ,M\right)|\mu_{q^{\prime }},\sigma_{q^{\prime }}^{2}\right) & =E\{\gamma \left({\tilde{x}}_{n}\left(\tau ,M;q\right)|\mu_{q^{\prime }},\sigma_{q^{\prime }}^{2}\right)\} \end{array}$$
(8b)$$\begin{array}{cc} p\left({\bar{x}}_{n}\left(\tau ,M;q^{\prime }\right)\right) & =E\{p\left({\tilde{x}}_{n}\left(\tau ,M;q\right)={\bar{x}}_{n}\left(\tau ,M;q^{\prime }\right)\right),\forall q\} \end{array}$$
where $p\left({\tilde{x}}_{n}\left(\tau ,M;q\right)={\bar{x}}_{n}\left(\tau ,M;q^{\prime }\right)\right)$ is the probability that a symbol belongs to every element of the dictionary, $p\left({\tilde{x}}_{n}\left(\tau ,M;q\right)={\bar{x}}_{n}\left(\tau ,M;q^{\prime }\right)\right)=\gamma \left({\tilde{x}}_{n}\left(\tau ,M;q\right),{\bar{x}}_{n}\left(\tau ,M;q^{\prime }\right)\right)$, being $\gamma \left(\cdot \right)$ a Gaussian similarity function, and $\sigma_{q^{\prime }}^{2}\in R,\mu_{q^{\prime }}\in R^{M}$ the moments computed, respectively, as below: $$\begin{array}{cc} \mu_{q^{\prime }} & =\sum_{\forall q}{\tilde{x}}_{n}\left(\tau ,M;q\right)p\left({\tilde{x}}_{n}\left(\tau ,M;q\right)={\bar{x}}_{n}\left(\tau ,M;q^{\prime }\right)\right)\\ \sigma_{q^{\prime }}^{2} & =\sum_{\forall q}{\left({\tilde{x}}_{n}\left(\tau ,M;q\right)-\mu_{q^{\prime }}\right)}^{⊤}\left({\tilde{x}}_{n}\left(\tau ,M;q\right)-\mu_{q^{\prime }}\right)p\left({\tilde{x}}_{n}\left(\tau ,M;q\right)={\bar{x}}_{n}\left(\tau ,M;q^{\prime }\right)\right) \end{array}$$

Therefore, the proposed Entropy-based estimator, termed *VQ-En*, builds the probabilistic priors by assessing the Gaussian similarity between the input and vector-quantized representations for dealing with the scarce statistics because of small code-vector sets (formed through the Euclidean distance), as detailed in Algorithm 1.
**Algorithm 1** Building of VQ stochastic patterns.1:**procedure**Vector Quatization in *X*  2:    Input: ${\tilde{x}}_{n}\left(\tau ,M;q\right),q\in \left[1,Q\right]$  3:    Initialize the reduced set ${\bar{X}}_{n}\left(\tau ,M\right)$, then ${\bar{x}}_{n}\left(\tau ,M;1\right)={\tilde{x}}_{n}\left(\tau ,M;1\right)$ 4:    **for**
$q\in \left[2,Q\right]$
**do** 5:        Compute the distance between ${\tilde{x}}_{n}\left(\tau ,M;q\right)$ and ${\bar{X}}_{n}\left(\tau ,M\right)$. $d\left({\tilde{x}}_{n}\left(\tau ,M;q\right),{\bar{X}}_{n}\left(\tau ,M\right)\right)=\left|\right|{\tilde{x}}_{n}\left(\tau ,M;q\right)-{\bar{x}}_{n}\left(\tau ,M;q^{\prime }\right){\left|\right|}_{2}^{2},q^{\prime }\in \left[1,Q^{\prime }\right]$  6:        **if**
$||d\left({\tilde{x}}_{n}\left(\tau ,M;q\right),{\bar{X}}_{n}\left(\tau ,M\right)\right)>\rho {\left|\right|}_{1}=Q^{\prime }$
**then** 7:           ${\bar{X}}_{n}\left(\tau ,M\right)\leftarrow {\tilde{x}}_{n}\left(\tau ,M;q\right)$ 8:           $Q^{\prime }=Q^{\prime }+1$ 9:        **end if** 10:    **end for** 11:**end procedure**

## 3. Experiments and Results

We validate the proposed *VQEnt* approach for estimation of event-related De/Synchronization using the following stages: (*i*) Tuning of Entropy-based estimators: short-time window, Embedding dimension, and tolerance. (*ii*) Estimation of time-series for Event-Related De/Synchronization, aiming to explore their interpretation ability, and (*iii*) Activation of the sensorimotor area in distinguishing between MI tasks. Of note, tuning and validation are carried out within the MI interval, that is, [2.5–4.5] s.

### 3.1. Parameter Tuning of Compared Entropy-Based Estimators

Generally, every parameter influences the Entropy-based assessments of ERD/S, but contributing differently to two main aspects of performance: discriminability and physiological interpretability. A first decisive parameter is a short-time window that must be adjusted to extract the EEG dynamics over time accurately [32]. Related to building the sample-based alphabets, we investigate the following values of τ reported in MI tasks [33,34]: τ∈{1,1.5,2} s with 90% overlapping. Further, we explore the importance of the complexity parameters on building the embedded alphabets: threshold tolerance ρ, measuring the regularity of pattern similarity, and the embedding value *M*. In terms of distinguishing between different MI tasks, we assess the parameter contribution, employing the bi-class accuracy that is computed by the Linear Discriminant Analysis algorithm under a 10-fold validation strategy. Thus, to generate the embedded alphabets, both complexity parameters (ρ and *M*) are heuristically established to reach the best classification rate. To this end, we search within the interval of embedding dimension, M={1,2,3} and tolerance ρ={0.05,0.1,0.2,0.3,0.4,0.5,0.6,0.7,0.8,0.9}.

Table 1 displays the accuracy performed by every tested subject. Note that for interpretability purposes, the individuals are ranked in decreasing order of the performed accuracy to rate the BCI literacy. So, a previous MI study defined the BCI-literacy threshold at 70% [35]. In the following, this level will be marked with dashed lines on the plots. So, we rank all subjects by the accuracy achieved by *SampleEnt*, as follows: BO9T, BO8T, BO3T, BO1T, BO5T, BO6T, BO7T, BO2T, and BO4T.

As seen, the value of τ=2 s provides the lowest accuracy regardless of the evaluated Entropy-based estimator. Though the statistical differences are not high to be significant between the small windows, the choice of the shortest window τ=1 s seems to be the best option since it gives the highest mean accuracy with lower dispersion. To strengthen this selection, we highlight the fact that five of the individuals reach the best performance in this window (see the underlined scores).

Besides, the comparison between estimators shows that *SampleEnt* and *FuzzyEnt* have similar accuracy, while *VQEnt* outperforms a bit with the benefit of supplying the lowest dispersion. Moreover, the majority of subjects perform the best result using the sample-based VQ Entropy.

For illustrating the parameter tuning, Table 2 displays the values fixed for each estimator to achieve the best individual classifier performance. In the case of quantized stochastic patterns, the value M=2 appears to be enough, while by adjusting ρ∼0.3 leads to accurate estimates of accuracy. The impact of the investigated dynamics becomes evident from Figure 2 that illustrates the parameter variability for the proposed *VQEnt*. For better visualization, the tested subjects are split into three groups due to the differentiable behavior reported for their brain activity dynamics evoked in practicing MI tasks [36]. As widely-known, therefore, the optimal parameter setting depends on the complexity measured for each subject group.

### 3.2. Interpretability of Time-courses Estimated for Event-Related De/Synchronization

To have a better understanding, Figure 3 presents the ERD/S time-series of the Entropy-based methods computed for the best individual of each group within the MI interval [2.5–4.5] s. All ERD/S time-courses are estimated for the representative sensorimotor channels (that is, C3 and C4) as a response to either performed MI task. For the sake of comparison, the top raw displays the corresponding ERD/S trajectories calculated by the variational percentage in EEG signal power, as described by Equation ([Disp-formula FD1a-entropy-22-00703]). In this case, each trajectory is averaged across the whole trial set, providing a resolution that is much bigger than the one resulted from the tested Entropy-based methods since Δt≪τ.

For the right-hand task, the Entropy time-series of the contralateral electrode, C3, starts decreasing from the maximal value at a time sample close to 2s (after the cue onset) and reaches the lowest point at 3s. Further, the MI brain response begins increasing. As expected, the Entropy of electrode C4 behaves with the same pattern for the left-hand task, as detailed in [37]. At the same time, the time-courses of the ipsilateral electrode (C4 for the right hand, C3 – left-hand) holds high values over the MI interval. Therefore, the ERD/S patterns performed by each evaluated Entropy-based estimator fulfills the MI paradigm. That is, the ERD/s evolves more firmly on the electrodes located contralaterally to the hand involved in each task when a subject imagines the movement of its right/ left hand.

Nonetheless, the de/synchronization model is more evident for τ=1 (solid line), but the responses weaken and tend to be smoother as the time window elongates. Furthermore, the ability to learn MI tasks also influences: the higher the BCI literacy, the more evident the ERD/S patterns. While the subject B08T (performing the best) presents brain responses with marked differences between tasks, the time-series set of BT07 (achieving a very low classifier accuracy) is far from being a synchronization pattern within the trial timing. This finding follows some clinical studies, evidencing that BCI-illiterate subjects manifest a lack in event-related desynchronization, which is of keen importance to perform MI tasks satisfactorily [38].

On the other hand, the averaged time-courses seem to be similar at each validating set-up (i.e., by fixing the same time window and BCI literacy), and therefore, explaining the proximity of accuracy provided by the Entropy-based estimators. Still, there are subtle differences between them. For investigating this aspect in more detail, the trial-wise relationship is calculated through the following distance of similarity [39]: $$d\left(n,n^{\prime }\right)=\exp\left(-||H\left\{X_{nm}\left(\cdot ,\cdot \right);\cdot \right\}-H\{X_{n^{\prime }m^{\prime }}\left(\cdot ,\cdot \right);\cdot {\}||}_{2}^{2}/\sigma_{X}^{2},\forall n,n^{\prime }\in N,\right)$$
where $\sigma_{X}$ is the variance averaged across the trial set for each validated Entropy-based estimator $m,m^{\prime }\in \left\{\mathrm{SampEnt},\mathrm{FuzzyEnt},\mathrm{VQEnt}\right\}$.

In the case of subjects with average rates of BCI literacy over 70%, the top and middle rows of Figure 4 display the connection matrix of similarity, calculated at τ=1 s, showing that the MI brain response of *SampEnt* and *FuzzyEnt* algorithms are very close in shape. However, the ERD/S time-courses performed by *VQEnt* differs from other estimators in all cases of τ. Otherwise, each Entropy-based method becomes more separate from others, as depicted in the button row for BT07 with BCI illiteracy. In terms of the performed MI tasks, the lower and upper triangular parts of the connectivity matrix hold very subtle distinctions in each one of representative channels (C3 and C4) and regardless of the employed Entropy-based estimator.

### 3.3. Statistical Analysis

Intending to evaluate the contrasted methods, we perform the non-parametric permutation test commonly used in evaluating different effect types of evoked responses in EEG applications [40]. To estimate the p-value, the Monte-Carlo permutation partitions are chosen by clustering of all adjacent time-samples that exhibit a similar difference. In each subject-based permutation, we cluster the spatial and temporal adjacency across the trial set, for a fixed value of p<0.02. Figure 5 depicts the obtained topographical plot of two representative individuals (literate subject *B08T* and illiterate *B01T*), showing the channels that hold discriminant information in performing the MI task, which are computed within five non-overlapped time windows of interest: before task ([0.5–1.5] s), during MI task ([2.5–3.5] s and [3.5–4.5] s) and at the trial timing end ([4.5–5.5] s and [5.5–6.5] s).

As expected, there is no information about the MI task in the interval before the stimulus. Instead, discriminant information is mostly localized within both MI segments, but the estimates have very changing behavior in [4.5–5.5]. Note that the discriminate information fades at the trial timing end when either subject is performing a break. In the case of B08T, the discriminating activity involves the Centro-lateral primary motor area, supplementary motor area, frontoparietal, and primary somatosensory area, that is, the regions typical in hand MI practicing [41]. *B01T* shows a weak contribution in those areas also, but excluding the critical frontoparietal region [42].

### 3.4. Contribution of Sensorimotor Area to Distinguishing between MI Tasks

Figure 6 displays the relevance of each sensorimotor channel that is computed as the Euclidean distance between the activities of labeled trials [43]. As seen in the top row, BT08 has high values of relevance in channels C3 and 18 (left hemisphere), as well as in C3 and 14 (right hemisphere), meaning that both regions contribute alike. At the same time, either channel belonging to the longitudinal fissure area produces a little contribution. The relevance sets provided by all Entropy-based estimators are very similar and agree with the MI paradigm. Nonetheless, the C3 electrode is weaker than the 14 one (left hemisphere). Figure 7 displays the time-courses of *VQEnt*-based ERD/S, showing the differences in IM responses between the ipsilateral channels. As seen, electrode 14 is more potent than the representative C3, while electrode 18 is more potent than C4. This situation can be explained because of the volume conduction effect of EEG signals, which hold a low spatial resolution, and thus, lead to inaccurate measures of brain activity [44]. In the case of B01T, the assessed relevance set is comparable to those obtained by BT08, as shown in the middle row. However, the contribution from the left-hemisphere channels (C3 and 14) is higher than provided by the right hemisphere (C4 and 18), suggesting a right-hand dominance [45].

Using the estimated Entropy-based ERD/S time-series, we investigate the increased activity of the sensorimotor area that is related to the motor imagery paradigm, assessing the electrode contribution (or relevance) in terms of distinguishing between the labels. Namely, the following channels are considered: left hemisphere (C3, 9, 14, and 15), right hemisphere (C4, 11, 18, and 17), as well as the longitudinal fissure area (10, 16).

In the case of BT07, the relevance set redistributes across the whole sensorimotor area, increasing in value at each electrode. Moreover, the contribution of longitudinal fissure area starts growing, though these electrodes are assumed to have very modest participation in motor imagery activation. Thus, this subject with low performance shows fewer prominent features than those who perform better, as already has been reported in similar cases [46].

One more aspect to consider is the resulting accuracy due to the assessed electrode contribution after using the estimated Entropy-based ERD/S time-courses. In this regard, two different scenarios are considered: a) Addition of the whole EEG channel set, b) Incorporation of just the sensorimotor channels. In either case, training is conducted by adding every channel ranked in decreasing order of relevance. As displayed inFigure 8a, the individuals B01T and B08T deliver high values of accuracy. Moreover, in both cases, the *VQEnt* estimator presents the most accurate outcomes at a lower amount of electrodes. A similar situation takes place with the individual B07T, for which our proposed method remarkably increases the accuracy in comparison to the other tested Entropy estimators. In the latter scenario, Figure 8b reveals that the reduced channel set directly involved with the MI paradigm is enough to discriminate between tasks, providing an accuracy similar to the performed by the whole electrode set.

Nevertheless, though the *VQEnt* estimator allows enhancing the performance of the best literate subjects, our proposal fails in the case of B07T. One factor that may account for this result is the volume conduction effect since it also may affect the Entropy-based estimators, as referred in [47]. A detailed analysis of the relevance performed in the all-channel scenario shows that this individual redistributes his values all over the excluded neighboring frontal area.

Another point to highlight is the influence of noise on the entropy calculation. Specifically, to address the volume conduction problem, we perform a Laplacian filter that improves the spatial resolution of EEG recordings, avoiding the influence of noise from neighboring channels [48]. Figure 9 shows the cases of the entropy computation of channel C3 with (and without) spatial filtration. As seen, the entropy calculated from the raw data (left) does not present any de/synchronization related to elicited neural responses regardless of the tasks (left hand/right hand). Instead, the Laplacian filter reduces the effect of noise coming from neighboring channels, making clear the changes related to the stimulation of motor imagination.

## 4. Discussion and Concluding Remarks

We present the Entropy-based method, termed *VQEnt*, for estimation of Event-Related De/Synchronization using a dynamic description through quantized stochastic patterns, aiming to improve discriminability and physiological interpretability of motor imagery tasks. The validating results, obtained on the widely used database, show that *VQEnt* outperforms others sample-based approaches while providing adequate interpretability in Motor Imagery tasks. The proposed method is sample-based and builds the probabilistic priors by assessing the Gaussian similarity between the input EEG measurements and their reduced vector-quantized representation. Nevertheless, the following aspects are to be considered:

*Parameter tuning of Entropy estimators*: A first decisive parameter is a short-time window that must be adjusted to extract the dynamics over time from MI data accurately. The value τ=1 is fixed that gives the highest mean accuracy with lower dispersion, providing similar performance for all tested Entropy-based estimators. This choice is reported to be generally appropriate for most time-series that have dynamics with rapidly decaying autocorrelation function.

Moreover, we explore the influence of complexity parameters (threshold tolerance ρ and embedding value *M*) on building the embedded alphabets. According to the complexity values fixed to achieve the best classifier performance of each individual, *SampEnt* and *FuzzyEnt* demand symbols with more elements to encode more precise the rapid dynamics because of the relatively small value tuned for τ=1. However, for dynamic systems that have long-range correlation, the choice of different delays can have a significant impact on the calculation of Sample-based algorithms, leading to inconsistencies in the pairwise evaluation of the relative complexity between time-series, as discussed in [49]. As a result, the similarity pattern count calculation in Equation (Equation 5) will necessitate more statistics, which are supplied with the trial set. Instead, to encode dynamics, *VQEnt* relies on a quantized version that yields alphabets with a high compression ratio, and therefore, requiring symbols with more extensive representations (M=2,3). In other words, each symbol encodes not one, but several neighboring samples.

Furthermore, the fact that *VQEnt* alphabets have a high compression ratio avoids a significant impact of noise on the time-series, and it reduces in complexity the choice of ρ. In contrast, the fuzzy and sample methods tend to be more susceptible to the noise effect, resulting in larger values of ρ. Overall, the parameter tuning of Entropy estimators depends on the BCI literacy rate. Of note, we test three Entropy-based methods that have as a significant advantage that they do not need any reference power value, which is far from being easy to adjust while highly influences the ERD/S estimation.compression ratio, and therefore, requiring symbols with more extensive representations (M=2,3). In other words, each symbol encodes not one, but several neighboring samples.

*Interpretability of estimated ERD/S time-series*: Generally, the ERD/S dynamics, performed by each considered Entropy-based estimators, fulfills the experimental paradigm of practiced MI tasks. However, due to inherent nonstationarity and a poor signal-to-noise ratio of EEG signals, location and amplitudes of brain activity sources have substantial variability in patterns across trials. For understanding the causes of inter- and intra-subject variability in performance, the database subjects split into groups with the differentiable behavior of brain dynamics in MI tasks. As a pivotal parameter, the short-time window is fixed to τ=1 to achieve a higher classifier accuracy. The first finding is that the complexity parameters that quantify the EEG data dynamics vary for each subject group, resulting in a differentiated optimal parameter setting. Moreover, the ability to learn MI tasks also influences, meaning that the higher the BCI literacy, the more consistent the ERD/S patterns of motor imagery. Besides, the connection matrix of similarity confirms that the ERD/S time-series performed by *VQEnt* are different in shape from the ones build by *SampEnt* and *FuzzyEnt* algorithms.

*Activation of the sensorimotor cortex during motor imagery*: We assess the contribution to distinguish between MI labels and prove that the relevance sets, provided by the left and right hemispheres, are similar despite the estimated Entropy-based ERD/S time-series. However, in individuals with the illiteracy rate, the relevance set spreads and increases abnormally across the whole sensorimotor area. As a result, literate individuals deliver high values of accuracy. Moreover, the *VQEnt* estimator presents the most accurate outcomes at a lower amount of electrodes so that reduced channel set directly involved with the MI paradigm is enough to discriminate between tasks, providing an accuracy similar to the performed by the whole electrode set.

Nonetheless, some issues remain to improve the effectiveness of the developed *VQEnt* approach for the estimation of ERD/S. Firstly, the extraction of VQ alphabets should be improved, by instance, using more elaborate distances for their construction. Moreover, it would be of benefit to incorporate other types of stochastic embedding to relax the parameter tuning of the used complexity representations. However, by increasing efficiency of the extracted symbols, the computational burden must also be examined. So far, the implementing cost of *VQEnt* exceeds more than 50% other sample-based algorithms. Also, the concept of illiteracy faces several pitfalls in BCI research so that alternative criteria should be considered [50].

As a concluding remark, we propose to enhance the entropy-based estimation by extracting more information about amplitudes of time-courses that show more differences in distinguishing between MI tasks. We hypothesize that by extracting a more reliable representation of the stochastic patterns, the discriminability of labeled tasks can be increased while preserving elicited brain neural activity’s physiological interpretation.

As future work, the authors plan to expand the developed Entropy-based method to introduce more information coming from neighboring channels to build the conditional probabilistic priors. We also intend to validate our proposal on databases that contain more subjects with a broader class of dynamics, aiming to understand why some subject groups show different performances in the same system.

## Figures and Tables

**Figure 1 entropy-22-00703-f001:**
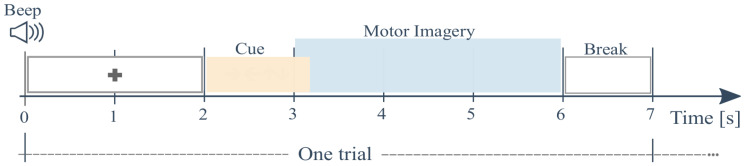
Paradigm trial timing of the validated MI database. The analysis is performed within the segment $T_{MI}$, including the start and termination of MI tasks.

**Figure 2 entropy-22-00703-f002:**
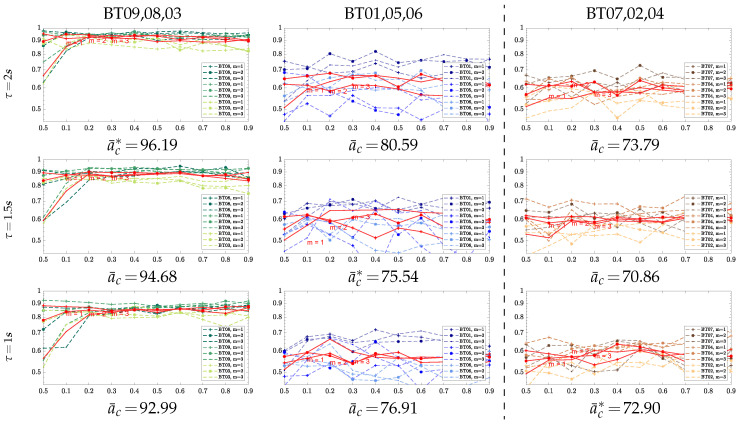
Performance variability depending on the individual parameter set-up of *VQ-En* estimator, accomplished at the examined windows τ. Presented values of accuracy ${\bar{a}}_{c}$ are averaged across the subjects belonging to each considered group.

**Figure 3 entropy-22-00703-f003:**
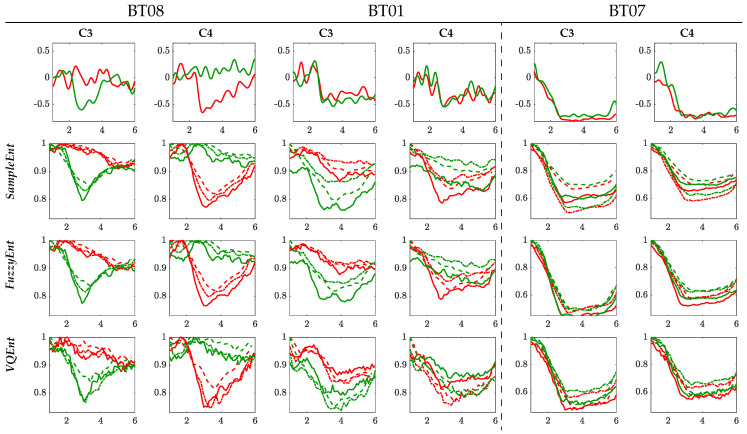
Individual ERD/S time-course of channels C3 and C4 performed by each tested Entropy-based estimator, averaging over all single trials for the right hand task (green color) and left hand (reed). Solid line τ=1 s, dash-dotted line τ=1.5 s, and dash line τ=2 s.

**Figure 4 entropy-22-00703-f004:**
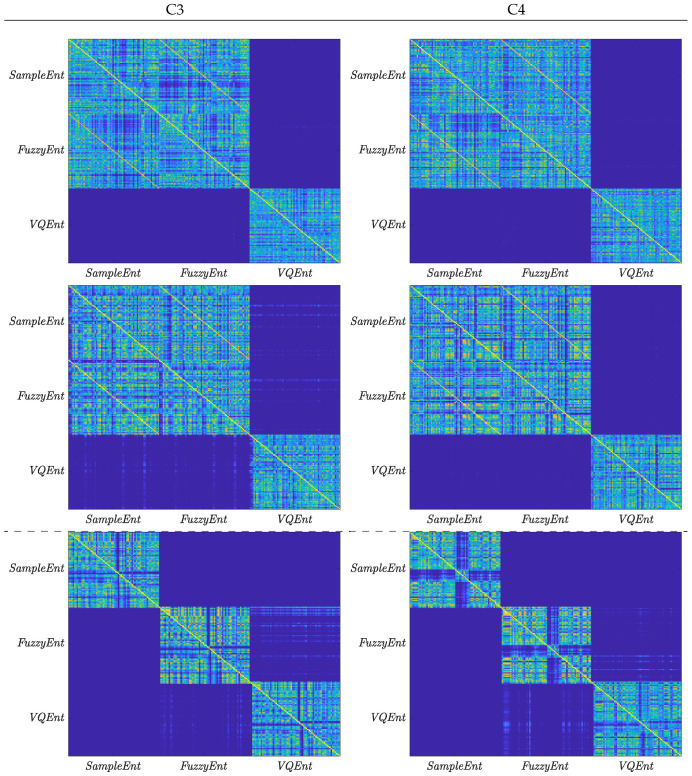
Asymmetric connection matrix of similarity between *SampEnt*, *FuzzyEnt*, and *VQEnt* performed by subjects with different rate of BCI literacy, and estimated across all trial set at τ=1 s. All entries above the main diagonal reflect the right label, while the lower triangular is for left label.

**Figure 5 entropy-22-00703-f005:**
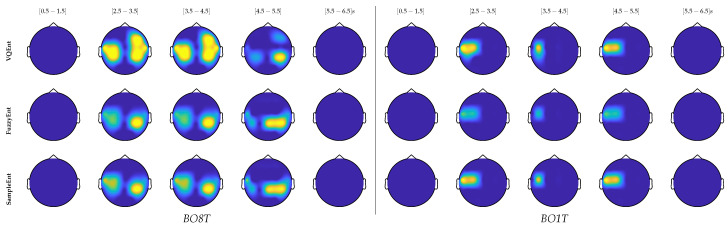
Statistical analysis of two representative individuals (literate subject *B08T* and illiterate *B01T*), showing the channels that hold discriminant information in performing the MI task.

**Figure 6 entropy-22-00703-f006:**
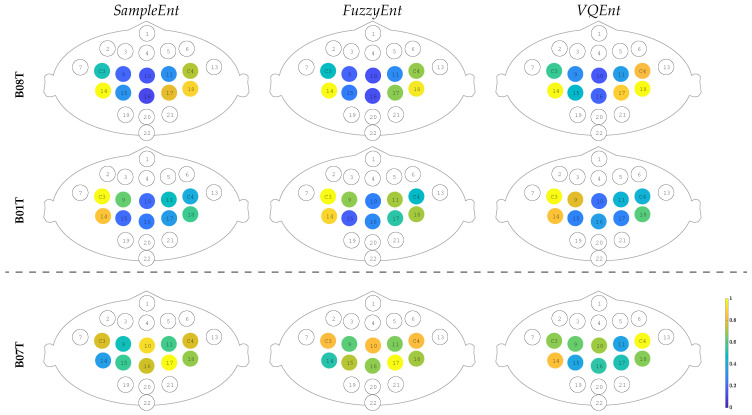
Sensorimotor electrode contribution in classifying MI tasks estimated through Entropy-based ERD/S time-series. Relevance weights of uncolored electrodes are not considered.

**Figure 7 entropy-22-00703-f007:**
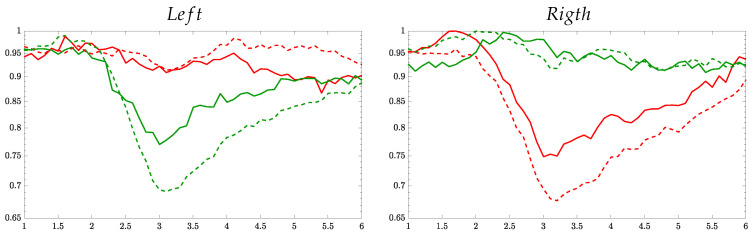
Detailed illustration of estimated ERD/S time-series: C3 vs ch14(dashed line), and C4 vs ch18(dashed line)

**Figure 8 entropy-22-00703-f008:**
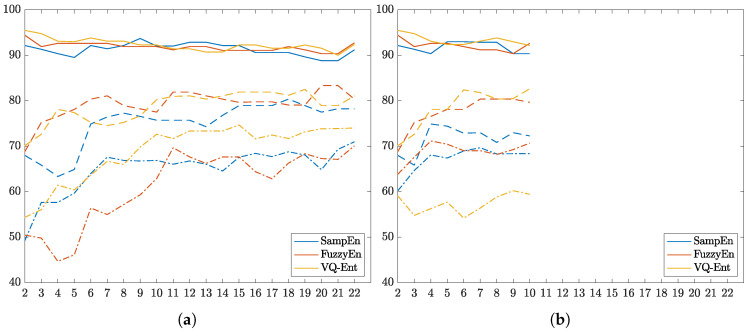
Classifier performance of subjects achieved by feeding each channel ranked by relevance. (**a**) Entropy-based relevance computed for all electrodes. (**b**) Entropy-based relevance of the sensorimotor electrodes.

**Figure 9 entropy-22-00703-f009:**
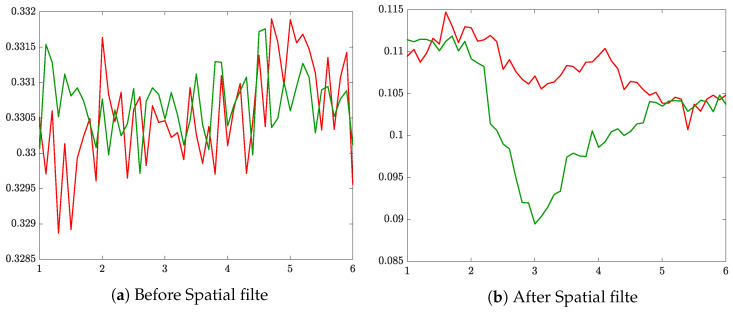
Example of Laplacian filtering to reduce the volume conduction effect on Entropy estimation.

**Table 1 entropy-22-00703-t001:** Influence of the short-time window on the bi-class classifier accuracy performed by each tested Entropy-based estimator. Notation ${}^{*}$ stands for the values of τ reaching the best accuracy of MI tasks. Note that individuals are ranked in decreasing order to rate the BCI literacy. The best individual scores are underlined while the best values performed between the estimators are marked in black.

#	*SampleEnt*			*FuzzyEnt*			*VQEnt*		
τ **[s]**	**2**	**1.5**	1 *	2	1.5 *	1	2	1.5	1 *
**B09T**	94.9 ± 8.3	95.7 ± 6.8	94.1 ± 5.21	94.2 ± 7.1	95.1 ± 7.2	95.0 ± 5.4	96.8 ± 5.2	96.6 ± 6.7	97.4 ± 4.0
**B08T**	94.4 ± 8.9	94.3 ± 8.3	92.0 ± 10.0	96.9 ± 3.8	96.1 ± 5.4	92.7 ± 8.7	97.6 ± 3.6	95.4 ± 6.2	92.4 ± 3.2
**B03T**	94.9 ± 3.4	91.3 ± 7.0	88.2 ± 6.4	89.7 ± 5.9	88.9 ± 6.8	86.1 ± 6.5	94.1 ± 5.4	92.0 ± 6.1	89.2 ± 8.6
**B01T**	81.2 ± 12.4	80.2 ± 14.7	78.2 ± 11.1	79.6 ± 11.1	81.1 ± 8.7	80.4 ± 11.6	81.9 ± 7.9	80.4 ± 9.2	81.1 ± 7.5
**B05T**	71.7 ± 11.4	73.7 ± 12.9	74.8 ± 12.4	73.0 ± 10.7	79.3 ± 6.9	75.5 ± 8.6	68.4 ± 10.2	71.4 ± 15.2	72.1 ± 10.3
**B06T**	70.3 ± 16.3	75.4 ± 12.8	72.9 ± 10.9	69.5 ± 11.2	73.9 ± 13.8	75.9 ± 6.2	69.6 ± 14.1	74.8 ± 12.2	77.5 ± 7.3
**B07T**	66.9 ± 11.9	67.7 ± 14.7	71.0 ± 10.7	67.8 ± 14.9	70.0 ± 14.3	70.1 ± 13.1	72.7 ± 13.8	71.9 ± 16.5	74 ± 10.1
**B02T**	59.4 ± 13.8	61.3 ± 8.7	68.5 ± 11.7	56.6 ± 7.7	60.9 ± 10.9	67.5 ± 16.8	65.7 ± 12.2	67.5 ± 11.0	73.5 ± 11
**B04T**	60.5 ± 11.8	62.1 ± 15.5	62.9 ± 11.0	58.1 ± 10.9	64.2 ± 6.5	65.1 ± 8.9	65.8 ± 14.3	73.2 ± 12.0	71.2 ± 10.7
*Mean*	77.1 ± 10.9	78.0 ± 11.3	78.0 ± 9.9	76.2 ± 9.3	78.8 ± 9.0	78.7 ± 9.5	**79.2 ± 9.7**	**80.4 ± 10.6**	**80.9 ± 8.1**

**Table 2 entropy-22-00703-t002:** Tuning of complexity values, threshold tolerance ρ and embedding value *M*), performed at τ=1 s, fixing Q=250-*M*. Notation $Q^{\prime }$ stands for the reduced size of VQ alphabets.

	*SampleEnt*	*FuzzyEnt*	*VQEnt*
#	M	ρ	M	ρ	M	ρ	$Q^{\prime }$
**B09T**	2	0.9	2	0.3	2	0.3	83
**B08T**	1	0.9	1	0.3	3	0.6	47
**B03T**	3	0.9	3	0.6	2	0.1	116
**B01T**	1	0.8	1	0.2	2	0.2	86
**B05T**	1	0.8	3	0.6	2	0.1	110
**B06T**	3	0.9	1	0.9	2	0.6	47
**B07T**	1	0.5	1	0.6	3	0.9	32
**B02T**	2	0.8	1	0.05	2	0.3	72
**B04T**	2	0.6	1	0.5	3	0.9	30
*Median*	1	0.8	2	0.5	2	0.3

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
