# Peer review of "Entropy-Based Estimation of Event-Related De/Synchronization in Motor Imagery Using Vector-Quantized Patterns"

_entropy, 2020, doi:10.3390/e22060703_

Round 1

Reviewer 1 Report

The motor imagery tasks are more and more popular in ivestigations of Brain-Computer Interfaces (BCIs) effectiveness and general properties. In this paper author undertake the problem of EEG event related De/Synchronization using entropic analysis. The use 22 electrode dataset and propose the algorithm for estimations of sensimotor activities during motor imaginery tasks. Their efforts allow to decrease the number of electrodes taken into consideration and to achieve same results as in the techniques konwn before. Interesting paper from thoe point of view of computational power consumption research. In order to set the article problems in the BCI literature better it could be adviced to search in WierzgaÅ‚a, Piotr, et al. "Most popular signal processing methods in motor-imagery BCI: a review and meta-analysis." Frontiers in neuroinformatics 12 (2018): 78.

Reviewer 2 Report

Summary and general comments

This study aimed at investigating an entropy-based method, named VQEnt, for estimation of ERD/S using quantized stochastic patterns as a symbolic space, to improve correspondent discriminability and physiological interpretability.

The authors claimed that reduced channel set directly involved with the MI paradigm is enough to discriminate between tasks, providing an accuracy similar to the performed by the whole electrode set.

As a whole, the method in the study has some novelty.  

However, there are several questions need to be answered.

Major comments

  1. The authors need to add statistical analysis methods to enhance results.
  2. Please state your hypothesis and aim(s) of the study clearly in the manuscript. Is the study designed (i.e. protocol) to test a hypothesis or fulfill your aims?

Is your study designed (i.e. protocol) to test the hypothesis or fulfill your aims? Do the conclusions that you reach completely answer your original questions?

  1. Why did not de-noise effects be included in the study for all methods in the manuscript?
  2. The representation of the experimental procedure for the testing subjects was not clearly presented? The authors need to show some information about the testing subjects. Age? Sex? Subject number? Healthy? Patients? Grouping? The Inclusion and Exclusion Criteria?

Minor comments

  1. The authors need to put Algorithm 1 in Table 1. In addition, the format of the other tables needs to be changed.
  2. The authors need to add a new section for statistical analysis.
  3. The abstract should be more precise.

Round 2

Reviewer 2 Report

The authors addressed the criticisms and remarks raised by me.